# Neuro-Inflammatory and Behavioral Changes Are Selectively Reversed by *Sceletium tortuosum* (Zembrin^®^) and Mesembrine in Male Rats Subjected to Unpredictable Chronic Mild Stress

**DOI:** 10.3390/cells14131029

**Published:** 2025-07-04

**Authors:** Johané Gericke, Stephan F. Steyn, Francois P. Viljoen, Brian H. Harvey

**Affiliations:** 1Center of Excellence for Pharmaceutical Sciences, Faculty of Health Sciences, North-West University, Potchefstroom 2531, South Africa; johane.gericke@gmail.com (J.G.); stephan.steyn@nwu.ac.za (S.F.S.); francois.viljoen@nwu.ac.za (F.P.V.); 2South African Medical Research Council Unit on Risk and Resilience in Mental Disorders, Department of Psychiatry and Neuroscience Institute, University of Cape Town, Cape Town 7505, South Africa; 3The Institute for Mental and Physical Health and Clinical Translation, School of Medicine, Deakin University, Geelong, VIC 3220, Australia

**Keywords:** anhedonia, antioxidant, anxious depression, hyperarousal, inflammation, phosphodiesterase 4B

## Abstract

*Sceletium tortuosum* (ST) induces antidepressant and anxiolytic effects, purportedly by monoamine regulation, anti-inflammatory and antioxidant properties, and phosphodiesterase 4 (PDE4) inhibition. These multimodal actions have not been demonstrated in an animal model of major depressive disorder. Wistar rats (both sexes) were subjected to 8-week unpredictable chronic mild stress, subsequently receiving saline, a standardized ST extract, Zembrin^®^ 25 and 12.5 mg/kg (ZEM25 and ZEM12.5), its primary alkaloid mesembrine (MES), or escitalopram (20 mg/kg) for 36 days. Sucrose preference, open field, Barnes maze, and forced swim tests were performed, with cortico-hippocampal monoamines, inflammatory and oxidative stress markers analyzed post-mortem. Male, but not female rats, presented with increased anhedonia and anxiety but not despair. Males presented with increased hippocampal PDE4B expression, increased dopamine metabolites, and decreased cortical serotonin. In males, ZEM12.5 decreased anhedonia- and anxiety-like behavior, decreased cortical and hippocampal PDE4B, and increased plasma interleukin-10. MES induced a transient decrease in anhedonia-like behavior and increased hippocampal serotonergic and cortical dopaminergic activity, whilst decreasing hippocampal PDE4B. ZEM25 increased plasma interleukin-10 but decreased cortical glutathione, indicating paradoxical anti-inflammatory and prooxidant effects. ZEM12.5 and MES more effectively addressed anxious–depressive-like behavior and stress-induced inflammation and monoaminergic alterations, respectively. Multitargeted actions on monoamines, redox-inflammation, and PDE4 may provide ST with antidepressant effects across multiple symptom domains, although mutually synergistic/antagonistic effects of constituent alkaloids should be considered.

## 1. Introduction

Major depressive disorder (MDD) represents a major component of disease burden [1], with a global prevalence of 10–20% [2]. Due to inadequate targeting of the pathological processes involved [3], current treatments are at best 50% effective [4]. The presence of multiple co-morbidities, such as cardiovascular and endocrine disorders as well as other psychiatric disorders [5], often complicates diagnosis and treatment. In this regard, anxiety and reward-based manifestations, e.g., anhedonia, anxiety, excitability [6,7,8], are not only common but have different biological underpinnings [3].

Animal models with robust translational validity are an important tool in drug discovery [3,6,7]. As with humans, appropriate housing conditions are critical for psychosocial wellbeing in rodents [7]. Poor housing conditions, like wet bedding, a lack of food and water, switching the lights on and off, light–dark cycle reversal, and noise, represent significant stressors for these animals [7]. Indeed, this lack of care engenders adverse behavioral manifestations akin to those observed in MDD [6,7,8].

The unpredictable chronic mild stress (UCMS) model has strong predictive and face validity with respect to the cumulative depressogenic and anxiogenic effects induced by chronic adversity, such as work stress, interpersonal conflicts, financial problems, isolation and loneliness, and/or loss of a loved one, especially if these persist over an extended period [9]. An example of this is the increased prevalence of MDD and anxiety experienced during and following the COVID-19 pandemic [10,11,12]. Therefore, adverse environmental experiences, as experienced in the UCMS rodent model, have strong translational validity for human mental health disorders, such as MDD [9], and therefore hold value as a preclinical drug discovery tool [7].

The specific pathophysiological mechanisms underlying the above-mentioned behavioral responses are generally ascribed to a dysregulated stress response. During acute stress, cortisol is released from the hypothalamic pituitary adrenal (HPA) axis to heighten arousal, improve and focus attention and cognition, accelerate motor responses, and mount specific metabolic and anti-inflammatory responses [13,14]. This process is under negative feedback control that maintains homeostasis [15,16]. However, during uncontrolled chronic stress, this negative feedback system is compromised [17], leading to long-term deleterious effects on the brain. The resulting allostatic load leads to reduced synaptic connectivity, hippocampal shrinkage and dysfunction, chronic neuro-inflammation, and monoamine deficits [18,19]. It is therefore not surprising that MDD invariably presents with hypercortisolemia [20] and shortfalls in monoaminergic function that contribute to the mentioned cognitive and socio-behavioral impairments [3].

Chronic inflammation activates the tryptophan–kynurenine pathway [21,22] and increases levels of the glutamate agonist, quinolinic acid, which in turn decreases the tryptophan necessary for serotonin (5-HT) synthesis. In addition, increased oxidative stress and mitochondrial damage increase pro-inflammatory cytokines [23], e.g., interferon-gamma (IFN-γ) and tumor necrosis factor-alpha (TNF-α), and the expression of phosphodiesterase 4 (PDE4), the latter hastening cyclic adenosine monophosphate (cAMP) breakdown [24]. Reduced cAMP impacts adenylate cyclase-dependent receptor signaling, e.g., 5-HT_1A_, thereby compromising serotonin and other monoamine receptor pathways [3]. Reduced cAMP also decreases anti-inflammatory and increases pro-inflammatory cytokine release, thereby worsening the neuro-inflammatory state [24,25]. Importantly, the inhibition of the cAMP pathway decreases brain-derived neurotrophic factor (BDNF) expression, important in synaptogenesis and neuroplasticity [24,26] and hypothesized to be involved in the neuro-progression of depressive symptomology [27]. PDE4 therefore plays a central role in the neuroimmune response to stress and underlies the complex pathophysiology of MDD.

The South African succulent plant, *Sceletium tortuosum* (ST), has shown potential antidepressant and anxiolytic activity, albeit in healthy populations [28]. ST is associated with various antidepressant-relevant mechanisms, including the upregulation of vesicular monoamine transporter (VMAT) 2 [28], the inhibition of the 5-HT transporter (SERT), monoamine oxidase (MAO), and PDE4 activity [29], as well as the modulation of glucocorticoid production [30]. Said pharmacological actions are attributed to the ST alkaloids, mesembrine (MES), mesembrenol, mesembranol, and mesembrenone, each of which possesses unique pharmacological properties and synergistic potential [28]. In this regard, MES alone may contribute significantly to the anxiolytic-like effects of ST, while its antidepressant-like activity is likely the result of pharmacological synergism [31].

While clinical and preclinical (in vivo and in vitro) studies have supported the antidepressant properties of ST [28], the lack of studies in patients diagnosed with MDD has increased the need for validated animal models to provide the necessary translational evidence and support [32]. Based on the overlap of the pathophysiological underpinnings of the UCMS rat model with MDD, this study investigates the antidepressant and/or anxiolytic effects of a standardized extract of ST, Zembrin^®^ (ZEM). Male and female Wistar rats were therefore subjected to a UCMS protocol [33,34], since including both sexes represents a more clinically relevant subject population [35]. UCMS does not appear to show a consistent sex bias, with studies reporting conflicting results [36,37,38,39]. With the hippocampus (HC) and frontal cortex (FC) critical in mood regulation and cognitive processing [40], monoamines, PDE4 expression, oxidative stress, and anti- and pro-inflammatory cytokines were assessed in these brain areas, along with various behavioral parameters akin to MDD. Since MES may be a more significant contributor to the anxiolytic effects of ZEM [31], MES and the selective SERT inhibitor (SSRI), escitalopram (ESC), were used as reference compounds.

Our working hypothesis was that a UCMS protocol would induce depressive-like behavior, in particular, despair, anhedonia-, and anxiety-like symptoms, and cognitive impairment in Wistar rats independent of sex. Moreover, simultaneous changes in brain PDE4, 5-HT, and dopamine (DA) concentrations, as well as inflammation and oxidative stress markers, would provide an important link to causation. These changes would furthermore be reversed by ZEM and MES administration through multitarget involvement, with biobehavioral effects being comparable or superior to ESC.

## 2. Materials and Methods

### 2.1. Plant Materials and Chemical Profiling

Zembrin^®^ (HG&H Pharmaceuticals, Johannesburg, South Africa; Lot: SCE0420-2003) is a standardized dry hydroalcoholic extract of the aerial parts of Sceletium (local names: kanna, channa, kougoed), also known as *Sceletium tortuosum* (L.) N.E. Br. (Syn. Mesembryanthemum tortuosum L. (http://www.worldfloraonline.org, accessed on 15 May 2025). It presents with a drug-to-extract ratio of 2:1 (*w/w*), with extraction solvents being water and alcohol, standardized to 0.40% total alkaloids. The chemical composition of the Zembrin^®^ preparation is standardized to mesembrenone plus mesembrenol, contributing greater than or equal to 60%, and mesembrine contributing 20% or less [28]. In addition, the isolated alkaloid of ST, namely mesembrine (Lot: ROM0921BRINE; 98% purity), was obtained from HG&H as noted above. The botanical material used to produce the extract was authenticated by a botanist (A. Viljoen), with a voucher specimen stored at the Tshwane University of Technology, Pretoria, South Africa. While the precise proportions of alkaloids within Zembrin^®^ are proprietary, the manufacturer is open to providing specific details for research purposes, contingent upon a non-disclosure agreement (NWU-HG&H Pharma Material Transfer Agreement).

### 2.2. Animals

Adult Wistar rats (*n* = 95; 48 females, 47 males) were bred, housed, and supplied by the PCDDP Vivarium (SAVC reg: FR15/13458) of the North-West University. Group sizes for the main analysis were determined with an A priori power analysis, set at an expected large effect (*F* = 0.4), α error probability of 5% and 80% power, and supported by previous studies [41,42,43]. Control group animals were housed in groups of 3 rats per cage (no mixed-sex cages), while UCMS animals were single-housed from PND43 to PND101. All animals were housed under standardized conditions of temperature (22 ± 2 °C), relative humidity (40–60%), positive air pressure (air exchange rate of 18/h), and full-spectrum light (12:12 h light/dark cycle; lights switched on at 06:00 and off at 18:00). Under conditions of UCMS, animals were subjected to light–dark reversal (see below). For environmental enrichment, animals were housed in polypropylene cages (380 × 380 × 230 mm) supplied with corncob bedding, polyvinyl chloride pipes, and standard nesting material. Standard rat pellets and water were supplied ad libitum, except during food and water deprivation (as part of the UCMS protocol). Animals were weighed and monitored for health or signs of distress daily (using a standard monitoring sheet), with the experimental period lasting from PND43 until PND101. One adverse event—an ESC male mortality (cause unknown)—was reported, and appropriate action was taken in accordance with the standard operating procedures of the PCDDP Vivarium. All experiments set out in the research protocol were approved by a NWU research committee and the NWU-AnimCare Research Ethics Committee (NHREC reg. number AREC-130913-015; approval number: NWU-00449-21-A5) of the NWU. All animal handling conformed to the South African National Standard (SANS) for the Care and Use of Animals for Scientific Purposes (SANS 10386:2008) [44]. Animals were maintained and procedures performed in accordance with the code of ethics in research, training, and testing of drugs in South Africa and complied with national legislation in accordance with ARRIVE guidelines [45].

### 2.3. Drug Treatment

Escitalopram oxalate (ESC; BLD Pharmatech Ltd., Shanghai, China. Lot: BD19428), Zembrin^®^ (ZEM), and mesembrine (MES) (HG&H Pharmaceuticals, Johannesburg, South Africa) were administered once daily from PND76 until 100. All drugs were dissolved in physiological saline (which also served as a control treatment) and administered daily via oral gavage in volumes not exceeding 0.4 mL. ESC was administered at a dose of 20 mg/kg [46], with the doses of ZEM and MES extrapolated from a prior study in zebrafish larvae using conversion calculations [31]. With the proportions of alkaloids contained in the ZEM chromatographic fingerprint being proprietary knowledge, the precise dose of MES is subject to a non-disclosure agreement. The dose of MES was calculated according to its proportional content in ZEM 12.5 mg/kg (data disclosed on request; NWU-HG&H Pharma Material Transfer Agreement).

### 2.4. Study Layout

As summarized in Figure 1, animals were randomly divided into the following treatment groups: saline control (Ctrl) and saline stress (UCMS) (*n* = 24 per group); UCMS + ESC (ESC) (*n* = 11—one male mortality); UCMS + ZEM12.5 (ZEM12.5), UCMS + ZEM25 (ZEM25); and UCMS + MES (MES) (*n* = 12 per group). Control animals remained group-housed throughout the study, while animals undergoing UCMS were single-housed from PND43 to PND101. To prevent any confounding litter-specific or genetic influences, animals born on the same day from different breeding pairs were divided into the different treatment groups using a randomizer program (https://www.randomizer.org). For the assessment of anhedonia-like behavior, all animals received sucrose preference training from PND48 to PND50. On PND50, the baseline sucrose preference test (SPT) was performed after 23 h of food and water deprivation, whereafter a six-week-long UCMS stress protocol was initiated. Intervention took place between PND64 and PND100, followed by behavioral assessments between PND92 and PND101. Behavioral analyses included anxiety assessment in the elevated plus maze (EPM; PND92), cognition and anxiety in the Barnes maze (BM; PND93-97), locomotor assessment in the open field test (OFT; PND99 and PND100), behavioral despair in the forced swim test (FST), and anhedonia in the SPT (PNDs 50, 64, 78, 92, and 101). Animals were euthanized by decapitation on PND101, when trunk blood and brain samples were collected. To determine the effect of the intervention, animals subjected to the UCMS protocol were treated with the test drugs and evaluated with respect to changes in selected bio-behavioral parameters related to depression, anhedonia, anxiety, inflammation, and oxidative stress.

### 2.5. The UCMS Protocol

The UCMS model is a valid paradigm used to induce anhedonia-like behavior in rats, along with other symptoms of MDD (i.e., despair, hopelessness, and cognitive impairment) while also inducing neurobiological changes akin to MD and anxiety [7,47]. Since a previous study in the Flinders Sensitive Line (FSL) rat suggested model-specific variability in the response to ZEM [32], the UCMS model was selected due to its validity across several symptom fields and MDD-related biological changes. While some studies indicate sex differences, there is a lack of consensus regarding the specific sex effects [36,37,38,39]. Keeping this in mind, a meta-analysis found that female rats are not more variable than males in neuroscience, which ultimately informed our decision to include both sexes in this study. That said, the UCMS model has been criticized for a lack of inter-laboratory reliability in behavioral response [48]. The method followed previously published protocols showing commonality in design and behavioral response [33,34].

Stressed groups were single-housed and exposed to UCMS for six weeks, starting on PND50. Stressors included a 45° cage tilt (4–8 h); the removal of bedding (4–8 h); wet cage interior for 2 to 4 h (1 cm water depth in the home cage without bedding); wet bedding (4–8 h); foreign object exposure (24 h; toy dinosaurs, hanging mirrors, toy plastic boats); white noise exposure (85 dB) (2 h); previously used cage swap with unknown rat from the same group and sex (24 h); pairing with stressed intruder of same sex (24 h); light–dark reversal (12–24 h); restraint stress (90 min, twice over 6 weeks); food and water deprivation for 23 h (only before SPT).

Animals were exposed to one stressor per day every day (except for four random “off” days) in an unpredictable order to prevent habituation [47]. Stressors were sometimes combined to increase the unpredictability of the stressor, e.g., cage swap with intruder exposure or white noise with cage tilt, among others. Animals had free access to food and water during stress (except before SPT and during restraint stress). All stressors were applied during the day in home cages, while stressors that would affect other animals housed in the facility were performed in a separate room.

### 2.6. Behavioral Tests

#### 2.6.1. Sucrose Preference Test (SPT)

The SPT is frequently used in rodent studies to assess anhedonia-like behavior, and most often accompanies UCMS due to impairments in reward processing [47]. Here, decreased sucrose consumption is used as a measure of reduced preference for a previously rewarding (hedonic) activity, thus akin to the anhedonia typically seen in MDD [8]. The assay focuses on the choice between sucrose and water and has been correlated with a decrease in reward sensitivity [47]. The SPT method was based on previous studies using the UCMS protocol [43,49,50].

Firstly, animals received sucrose preference training prior to the initiation of the UCMS protocol (Figure 1). Drinking water was initially supplied in two bottles per cage. On PND47, animals were introduced to both bottles containing a 1% (*w/v*) sucrose solution for 24 h. On PND48, the animals were exposed to two bottles, one containing sucrose and the other water, for 24 h. The next morning (PND49), food and water were removed and only returned on the day of testing (PND50), when the rats were allowed 1 h to freely choose between water or the sucrose solution. The bottles were weighed before and after the test to calculate the sucrose preference with the formula: *consumed sucrose (g)/(consumed sucrose + water (g)).* Higher values indicate increased sucrose preference and decreased anhedonia-like behavior. Control animals were also food and water deprived and separated for the hour-long test period to obtain more accurate individual data. This procedure was repeated on PNDs 64, 78, 92, and 101 to monitor the progression or potential reversal of anhedonia-like behavior over the duration of the study and following treatment.

#### 2.6.2. Elevated Plus Maze (EPM)

Anxiety is a common comorbid symptom of MDD [51]. The EPM is used as a measure of anxiety-like behavior in rodents and is reliant on the natural tendency of these animals to seek refuge in dark and enclosed spaces to avoid exposure and/or predation [52]. The EPM took place from 18:00 on PND92 using a previously described method [53]. Briefly, the rats were placed on a plus-shaped maze with opposing open (50 (*l*) × 10 (*w*) cm) and black closed (50 (*l*) × 10 (*w*) × 50 (*h*) cm) arms, elevated 50 cm above the ground. Rats were placed in the center of the maze and left to explore for 5 min. Total distance moved, duration spent in the closed versus open arms, and entries into the arms (counted when the center-point of the body entered the arms) were recorded with EthoVision^®^ XT 16 software (Noldus^®^ Information Technology, Wageningen, The Netherlands). Entries into and time spent in the closed arms were interpreted as anxiety-like behavior.

#### 2.6.3. Barnes Maze (BM)

Cognitive deficits are a common comorbid symptom of MDD [16]. The BM assesses spatial memory and cognitive performance in rodents [54]. Anxiety can also be assessed since the animals may present with different behaviors indicative of anxiety, such as changes in locomotor activity and search strategies [55,56]. The BM protocol was based on a combination of previously described methods [57,58]. The setup included a round light-grey table (122 cm diameter, elevated 90 cm off the ground) with 20 holes (10 cm diameter each) spaced evenly around the edge of the maze. A black escape box was inserted underneath one of the holes to not be easily visible from the surface. Three floodlights (1000 lux total), a video camera, and a wireless speaker generating white noise (85 decibels) were mounted above the maze. Black curtains surrounded the maze, with eight colorful shapes attached to the curtain serving as visual cues. On PND93, animals were habituated to the maze for 1 min, without white noise or visual cues. If the escape box was not found after 1 min, the rat was gently prompted into the box, the hole covered, and allowed to remain there for 30 s. During the next three consecutive nights (PND94 to 96), rats were exposed to three acquisition trials of 3 min each, separated by 15 min intervals. The maze was thoroughly cleaned with soapy water between trials to remove odor trails. Rats were placed in the center of the maze and covered with a polystyrene box to allow them to self-orient in a random direction for 10 s to prevent experimenter-induced bias. The box was lifted, whereafter the timer and white noise were immediately started. The rat was allowed to explore the maze and locate the escape box using spatial mapping with the help of visual cues. The trial ended either after the rat entered the escape box or after 3 min. On PND97, a 90 s probe trial (no escape box added) was conducted. Parameters measured using EthoVision^®^ XT 16 tracking software (Noldus^®^ Information Technology, Wageningen, The Netherlands) included time spent in the probe zone (“pizza slice” area from the center to two holes to the left and right of the escape box) and time spent in the outer zone (2 cm border on the edge of the table, and 2 cm diameter around the table edge) versus the inner zone (holes and center), while manual scoring was used to determine primary latency (time to find the target hole), primary error rate (number of incorrect holes visited), as well as edge exploration (head dips in the edge zone). Longer latencies and more errors are regarded as spatial memory impairment.

#### 2.6.4. Open Field Test (OFT)

Psychomotor retardation or agitation are common symptoms of MDD [40]. The OFT is used to assess locomotor activity in rodents. The test is also useful to control for locomotor activity as a possible confounder in other behavioral tests, such as the elevated plus maze and forced swim test. In addition, the OFT is used to assess exploration and anxiety-like behavior in rodents [59,60]. The test was performed during the dark phases of PND99 and 100, respectively, as described previously [61]. Total distance moved was used as an indicator of general locomotor activity and scored in the second trial. Animals were individually placed in a black open field box (100 (*l*) × 100 (*w*) × 50 (*h*) cm) and allowed 5 min to explore. To increase the anxiogenic environment of the test, behavior was recorded under white light [62,63]. Locomotor activity was recorded with a ceiling-mounted camera and tracked and analyzed with EthoVision^®^ XT 16 software (Noldus^®^ Information Technology, Wageningen, The Netherlands).

#### 2.6.5. Forced Swim Test (FST)

Despair and failure to cope with adversity are common symptoms of MDD [16,40]. The FST assesses behavioral despair in rodents as well as coping strategies in an adverse environment [64]. Following a previously published method [61], rats were individually placed in a clear Plexiglas cylinder containing water (diameter 18 cm; depth 30 cm; ambient room temperature) under full-spectrum white light. On PND99, 30 min after the OFT, rats were exposed to a 15 min pre-swim session, followed by re-exposure to swim stress (test session) 24 h later (PND100). Behavior was recorded for 7 min, with the first and last minute discarded to avoid possible confounding influence of bubbles caught in the fur at the start of the session and possible exhaustion, respectively, on floating behavior. To prevent experimenter bias, the remaining footage was scored manually by an independent investigator using numbers generated by randomization software (https://www.randomizer.org/). Total time spent swimming (horizontal movement with quadrant crossing) and struggling (upward-directed movement against the sides of the cylinder) were used to score coping behavior, with immobility time (floating with minimal movement necessary to keep the head above water) used as a measure of behavioral despair.

### 2.7. Neurochemical Measures

#### 2.7.1. Sample Collection and Preparation

After decapitation on PND101, the FC and HC were dissected as previously described [65]. The FC and HC were pre-split into aliquots (right and left HC/FC) and transferred to 1.5 mL Eppendorf^®^ tubes, whereafter they were immediately snap frozen in liquid nitrogen and stored at −80 °C until the day of analysis. On that day, samples were thawed on ice before starting the various sample preparations for the different methods described below.

#### 2.7.2. Phosphodiesterase 4B, IL-10, and TNF-α

PDE4B concentrations in the HC and FC and plasma IL-10 and TNF-α levels were measured using an enzyme-linked immunosorbent assay according to the manufacturer’s instructions (PDE4B ELISA, Cloud-Clone Corp., Catalogue No: SEF642Ra, 96T); IL-10 ELISA, Elabscience^®^, Catalogue No: E-EL-R0016, 96T; TNF-α ELISA, Elabscience^®^, Catalogue No: E-EL-R2856, 96T). All results were derived from a standard curve and expressed as ng/mL.

#### 2.7.3. Monoamines

Quantitative analysis of FC and HC monoamines (5-HT, NA, and DA) and their respective metabolites, 5-hydroxyindoleacetic acid (5-HIAA) and dihydroxyphenyl acetic acid (DOPAC), were performed using high-performance liquid chromatography with electrochemical detection (HPLC-ECD), as previously validated [66]. Metabolites for NA were below the limit of detection. Monoamine concentrations are expressed as ng/mg wet weight brain tissue.

#### 2.7.4. Tyrosine, 3-Chlorotyrosine, GSH, and GSSG

Tyrosine (TYR), 3-chlorotyrosine (3-CLT), and glutathione in its reduced (GSH) and oxidized (GSSG) forms were measured using HPLC-ECD, as described previously [67,68,69]. Sample preparation was performed using the method as described for monoamine analysis (Section 2.7.3) [66]. Hereafter, 200 µL of the prepared sample was transferred to HPLC inserts in vials. Chromeleon^TM^ 7 Chromatography Data System version 7.3. software was programmed to inject 25 µL of both the standards and samples. A Venusil ASB C18 column, 4.6 × 250 mm, 5 μm, 300Å, with a mobile phase consisting of 60 µM NaH_2_O_4_P and 1% acetonitrile (pH = 2.80), was utilized for the separation of the analytes of interest on the HPLC-ECD system. A constant flow rate of 0.350 mL/min was set for separation. Standard solutions were prepared by dissolving approximately 1 mg each of tyrosine and 3-CLT in 10 mL of distilled water in amber volumetric flasks. From the stock solutions of each analyte, a 7-concentration range standard calibration curve was constructed to determine the linear range of each analyte. Please refer to Appendix A for the detailed method.

### 2.8. Statistical Analyses

GraphPad Prism Version 8.4 was used for all statistical analyses and graphical presentation. All data sets were screened for outliers using Grubbs’ test (*α* = 0.5), with identified outliers only removed in extreme cases. Due to the general lack of normal distribution, all data sets were analyzed using non-parametric tests. For model validation (CTRL vs. UCMS) and sex comparisons, the Mann–Whitney *U*-test was used, whilst the Kruskal–Wallis and Dunn’s multiple comparison tests were used to compare the intervention groups (ESC, ZEM12.5, ZEM25, MES) to the UCMS placebo control group. For the SPT, two-way repeated measures ANOVA was used (sphericity assumed), with multiple comparison results reported as Bonferroni-adjusted values. Where correlations between two variables were suspected, Spearman’s rank correlation was used for confirmation, whereafter only statistically significant findings were reported. Sample sizes were initially based on previous UCMS studies and a 5% probability of chance (see Section 2.2), with the main hypothesis and statistical power analysis centering around no sex differences, as mentioned in Section 2.5. However, UCMS was found to induce pronounced sex-specific differences across behavioral and biological domains. Male and female rats were subsequently grouped and analyzed separately. Considering the ensuing reduced *n*-values, a post-hoc power analysis indicated that at least 25% power was achieved, which would adversely impact the statistical power of the study. On these grounds, statistical significance was revised and set at *p* < 0.10. This approach is commonly employed in studies with small sample sizes to detect potentially meaningful effects that might otherwise be missed due to low power and thus mitigate the risk of Type II errors [70]. Moreover, an obligatory Cohen’s *d*-calculation was performed to increase the robustness of the data, with ≥0.8 accepted to indicate a large effect size. Cohen’s *d* is used to determine the practical significance of results independent of sample size, which can limit the false interpretation of sample-size-dependent *p*-values [70,71]. All in-text values are reported as mean values ± standard deviation, with graphical representations presented as the mean with a 95% confidence interval (CI).

## 3. Results

Only the most notable data are presented in the figures, although for reasons of transparency, all data are included in Appendix B. Thus, while the OFT and BM data are included in Figure 2 and Figure 4, the EPM data were omitted due to a lack of significant findings (except where noted).

### 3.1. Validation of the Model

#### 3.1.1. Effects of UCMS on Behavior (Figure 2; Table A1 in Section A.1)

As summarized in Table A1, baseline sucrose preferences were comparable between control and UCMS rats, irrespective of sex (*p* > 0.10). Moreover, although sucrose preference at PND101 was also similar (*p* > 0.10) between these groups, only the positive regression line slope of UCMS female rats was statistically non-zero (UCMS F; Figure 2A; F_1, 58_ = 6.65, *p* = 0.01; R^2^ = 0.10), whilst the apparent negative slope of the UCMS male rats did not reach statistical significance (UCMS M; Figure 2A; F_1, 57_ = 0.55, *p* = 0.46, R^2^ = 0.01). Despite the opposing directions, the slopes of UCMS male and female rats were statistically comparable (Figure 2A; F_1, 115_ = 3.19, *p* = 0.08), as were the slopes of the control and UCMS groups, with (Figure 2A; F_1, 235_ = 0.26, *p* = 0.60) and without considering the influence of sex (*p* > 0.10). Importantly, UCMS-exposed male rats had a lower mean sucrose preference (0.64 ± 0.05) over the entire experimental period compared to control male rats (0.83 ± 0.03; U = 969.5, *p* < 0.001, d = 0.90 [0.6; 1.3]) and UCMS-exposed female rats (0.82 ± 0.06; U = 1428, *p* = 0.05, d = 0.40 [0.1; 0.8]). Specific time-point comparisons are summarized in Table A1, with the most notable being the reduced sucrose preference on PND64 (U = 34, *p* = 0.03, d = 0.89; [1.8; 0.1]), PND78 (U = 19, *p* = 0.001, d = 1.16 [2.1; 0.3]) and PND92 (U = 39, *p* = 0.1, d = 0.87 [1.8; 0.03]) between UCMS-exposed and UCMS-naïve male rats.

None of the EPM parameters were statistically altered by UCMS, relative to their appropriate controls (Table A1). On PND100, however, distance moved in the OFT (Figure 2E) was higher in the UCMS male (U = 16, *p* = 0.0007, d = 1.55 [0.7; 2.5]) and combined (U = 136, *p* = 0.001, d = 1.08 [0.5; 1.7]) groups, when compared to their respective controls.

Irrespective of sex, UCMS-exposed animals were less immobile (Figure 2B; U = 110, *p* = 0.0001, d = 1.17 [1.8; 0.6]) and spent more time swimming (Figure 2C; U = 118, *p* = 0.0003, d = 1.09 [0.5; 1.7]) in the FST than their UCMS-naïve controls. UCMS-exposed male (U = 25, *p* = 0.006, d = 1.2 [2.1; 0.4]) and female (U = 26, *p* = 0.007, d = 1.06 [2.0; 0.2]) rats also spent less time immobile in the FST than their respective controls, whilst both UCMS-exposed male (U = 18, *p* = 0.001, d = 1.28 [0.4; 2.2]) and female (U = 40, *p* = 0.07, d = 0.81 [0.01; 1.7]) rats spent more time swimming. Contrastingly, only UCMS-exposed female rats spent more time struggling (Figure 2D; U = 36, *p* = 0.07, d = 0.98 [0.1; 1.9]) than their control counterparts.

Irrespective of sex, UCMS-exposed animals spent less time in the outer zone of the BM (U = 163, *p* = 0.009, d = 0.11 [0.7; 0.5]), compared to control animals (Figure 2F). This was also true for male (U = 27, *p* = 0.008, d = 0.81 [−1.7; 0.01]) but not female (U = 49, *p* = 0.20, d = 0.16 [−0.6; 1.0]) rats. Interestingly, UCMS-exposed male rats also made fewer primary errors (U = 17.5, *p* = 0.007, d = 1.14 [2.1; 0.2]), relative to unstressed controls (Table A1). In this regard, 25% (*n* = 3) of UCMS-exposed animals made no errors in finding the escape box, whereas 100% of the unstressed control animals erred in their exploration (Table A1). Finally, female UCMS-exposed rats explored the edge of the maze (head dips off the edge) less (U = 25.5, *p* = 0.005, d = 1.01 [2.1; 0.4]) than their unstressed female controls.

#### 3.1.2. Effects of UCMS on Neurochemical Markers (Figure 3; Table A2 in Section A.1)

Ignoring the influence of sex differences, UCMS only increased cortical tyrosine turnover (3-CLT/TYR) (Figure 3C; U = 138, *p* = 0.003, d = 0.77 [0.2; 1.4]), relative to stress-naïve controls. In male rats, UCMS increased hippocampal PDE4B concentrations (Figure 3A; U = 17, *p* = 0.01, d = 1.18 [0.3; 2.2]) and also increased cortical DOPAC concentrations (Figure 3E; U = 35, *p* = 0.06, d = 0.90 [0.02; 1.7]) and decreased cortical 5-HT concentrations (Figure 3D; U = 31, *p* = 0.06, d = 0.94 [1.9; 0.1]). Of note, a linear regression model (F_1, 97_ = 14.64, *p* = 0.0002, R^2^ = 0.13) suggested that hippocampal PDE4B concentration accounts for 13% of the observed mean sucrose preference variation and was consequently supported by a negative correlation between these factors (r(18) = −0.65, *p* = 0.002). Neither serotonergic nor dopaminergic turnover was affected by UCMS in any of the brain areas in male rats (Table A2). In females, UCMS only decreased cortical DOPAC concentrations (Figure 3E; U = 23, *p* = 0.004, d = 1.04 [1.9; 0.2]). As summarized in Table A2, UCMS had no significant effects on NA, IL-10, TNF-α, GSH, or GSSG in any of the groups (*p* > 0.10).

**Figure 3 cells-14-01029-f003:**
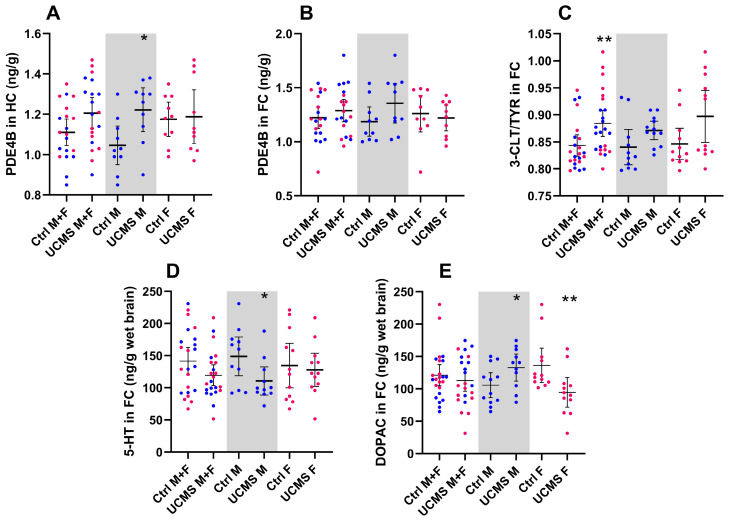
Neurochemical data for model validation in rats: (**A**) PDE4B levels in the HC and (**B**) FC; (**C**) 3CLT/TYR in the FC; (**D**) 5-HT levels in FC; (**E**) DOPAC levels in the FC of the Ctrl and UCMS groups of the combined sex group presented next to those of the male and female rats alone. Data points represent the mean ± 95% CI, with male and female rats indicated in blue and pink, respectively. Statistical analyses are presented in-text and summarized in Table A2. Significance (* *p* < 0.05; ** *p* < 0.01) is indicated above the UCMS group of the specific validation group.

### 3.2. Treatment Response and Influence of Sex

As elaborated on later, model validation indicated that UCMS failed to induce depressive (anhedonia, cognitive) and anxiety-like behavioral changes in the male and female combined group. To determine whether the impact of UCMS was similar in both sexes, the respective mean effect sizes across all bio-behavioral parameters were calculated for male and female rats. The overall effect of UCMS in male rats (d = 0.56 ± 0.40, CoV = 71.2%) was larger than that in females (d = 0.38 ± 0.30, CoV = 74.7%). In fact, that this difference also reached statistical significance (U = 1123, *p* = 0.02) further suggests that male rats may be more sensitive to UCMS, compared to their female counterparts. Because of the larger mean effect (d = 0.56 ± 0.40) observed in males, only the treatment intervention effects in male rats are discussed below. Consequently, this led to smaller sample sizes, which prompted the use of a less stringent *p*-value (*p* < 0.10) (see Section 2.8 for details). Data relating to the female rats and their response to UCMS are discussed in the Results and Discussion sections below, with other findings and sex comparisons described in Appendix A.

#### 3.2.1. Treatment Effects on Behavior (Figure 4; Table A3 in Appendix B)

There were no baseline (PND50) differences in sucrose preference (X^2^(4) = 6.27, *p* = 0.18) between male rats from the different experimental groups (Table A3), while UCMS did decrease sucrose preference over time in the male rats (Section 3.1.1). However, when comparing the therapeutic effects over time (all groups; Figure 4A), linear regression comparison indicates that although the slopes were similar across the experimental groups (F_5, 222_ = 1.71, *p* = 0.13), only male rats receiving ZEM12.5 (F_1, 28_ = 12.06, *p* = 0.002, R^2^ = 0.30), but not those receiving ESC (F_1, 23_ = 2.25, *p* = 0.15, R^2^ = 0.09), increased over time compared to UCMS placebo controls. Indeed, the ZEM12.5-induced increase observed between PND101 and baseline reached statistical significance (*p* = 0.03, d = 1.81 [0.5; 3.4]). Other significant time-specific differences were observed in the MES group, which displayed increased sucrose preference on PND64 (*p* = 0.006, d = 1.25 [0.2; 2.4]) and PND78 (*p* = 0.0003, d = 1.32 [0.3; 2.5]) (all data summarized in Table A3) compared to UCMS-exposed male rats. However, these effects were lost over time (PND92: *p* > 0.99, d = 0.67 [−0.3; 1.7]; PND101: *p* > 0.99, d = 0.43 [−0.6; 1.4]). ZEM12.5 was able to reverse decreased sucrose preference observed in UCMS male rats compared to controls on PND101 (*p* = 0.02, d = 1.13 [0.1; 2.2]. Finally, it is worth noting that male rats receiving either ZEM12.5 (0.81 ± 0.08) or MES (0.82 ± 0.08) displayed higher average sucrose preference over the five time points relative to the UCMS control group (0.64 ± 0.05).

**Figure 4 cells-14-01029-f004:**
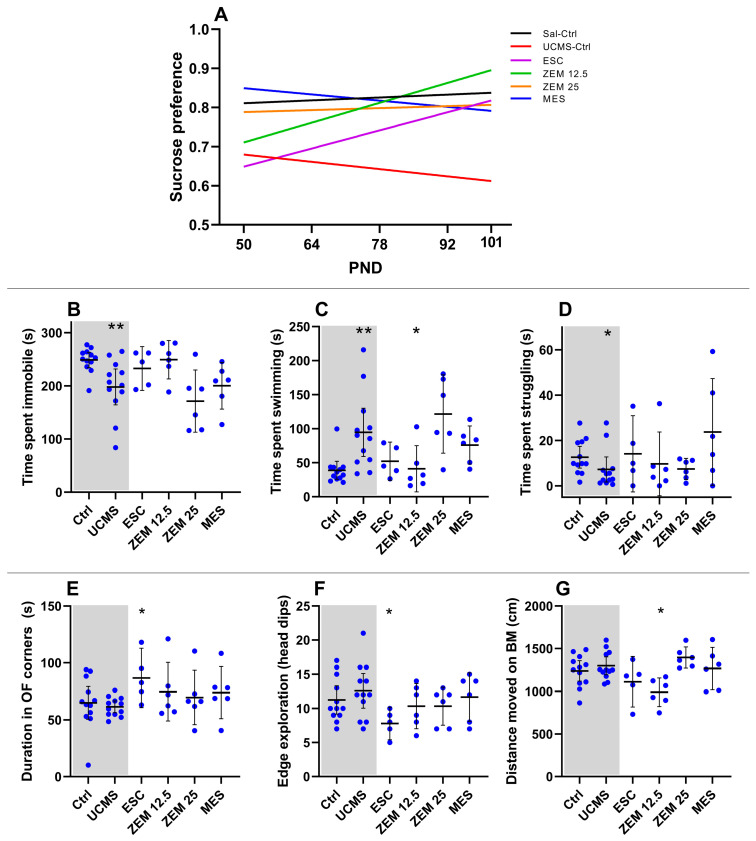
Treatment effects on behavior in male rats: Effects of the different treatments on (**A**) sucrose preference over time (PND—postnatal day); (**B**) time spent immobile in the FST; (**C**) time swimming in the FST; (**D**) time struggling in the FST; (**E**) time (s) in corners of the OFT; (**F**) edge exploration/head dips in the BM; (**G**) distance moved (cm) in the BM. * *p* < 0.1; ** *p* < 0.01. Data points represent the mean ± 95% CI. Statistical analyses are presented in-text and summarized in Table A3.

As mentioned in Section 3.1.1, UCMS increased mobility in male rats. Despite the significant influence of treatment on the time spent swimming in the FST, only ZEM12.5 decreased swimming (*p* = 0.06, d = 1.03 [2.1; 0.02]) compared to UCMS controls (Figure 4C). Similarly, although none of the treatments induced significant effects on immobility (Figure 4B; X^2^(4) = 8.44, *p* = 0.08) and struggling time (Figure 4D; X^2^(4) = 3.82, *p* = 0.43) in the FST, ZEM12.5-administration induced a large effect size increase in immobility (*p* = 0.17, d = 1.02 [0.01; 2.1]) versus UCMS controls. However, none of these parameters were altered in UCMS male rats (see Section 3.1.1).

UCMS increased locomotor activity in the OFT and decreased the time spent in the outer compared to the inner section of the BM (Section 3.1.1). Regarding these anxiety-like behaviors (Figure 4 and Table A3), only ESC increased the time spent in the corners in the OFT (Figure 4E and Table A3; *p* = 0.04, d = 1.85 [0.7; 3.2]) and decreased edge exploration in the BM (Figure 4F and Table A3; *p* = 0.06, d = 1.28 [2.5; 0.2] compared to UCMS placebo animals. None of the treatment groups showed any differences in distance moved in the OFT (Table A3). However, ZEM12.5 decreased distance moved in the BM (Figure 4G and Table A3; *p* = 0.02, d = 1.81 [3.1; 0.7]) compared to UCMS placebo animals.

#### 3.2.2. Treatment Effects on Neurochemical Markers (Figure 5; Table A4 in Appendix B)

As described in Section 3.1.2, UCMS increased hippocampal PDE4B and cortical DOPAC and decreased cortical 5-HT concentrations in the male rats but had no other significant effects on any of the other biomarkers. Experimental treatment interventions induced statistically significant changes in hippocampal PDE4B (X^2^(4) = 16.18, *p* = 0.003), 5-HT (X^2^(4) = 18.27, *p* = 0.001), 5-HIAA/5-HT (X^2^(4) = 14.56, *p* = 0.006), cortical PDE4B (X^2^(4) = 13.60, *p* = 0.009), GSH (X^2^(4) = 9.98, *p* = 0.04), and plasma IL-10 (X^2^(4) = 17.24, *p* = 0.002). Relative to UCMS placebo animals, MES increased hippocampal 5-HT (Figure 5C; *p* = 0.0009, d = 1.58 [0.5; 2.8]), decreased cortical GSSG (Table A4; *p* = 0.07, d = 1.21 [2.3; 0.2]), and decreased the hippocampal 5-HIAA/5-HT ratio (Figure 5D; *p* = 0.008, d = 2.26 [3.6; 1.1]) and PDE4B concentrations (Figure 5A; *p* = 0.04, d = 1.74 [3.1; 0.5]). As for the escalating doses of ZEM, ZEM12.5 decreased cortical (Figure 5B; *p* = 0.002, d = 2.24 [3.8; 0.9]) and hippocampal (Figure 5A; *p* = 0.0007, d = 2.5 [4.1; 1.1]) PDE4B concentrations whilst increasing plasma IL-10 levels (Figure 5G; *p* = 0.0007, d = 2.9 [1.5; 4.5]). On the other hand, ZEM25 decreased cortical GSH (Table A4; *p* = 0.03, d = 1.24 [2.4; 0.2]) and increased plasma IL-10 levels (Figure 5G; *p* = 0.03, d = 1.89 [0.7; 3.3]). The influence of the different treatment strategies was also statistically significant for plasma TNF-α (X^2^(4) = 10.37, *p* = 0.03), cortical (X^2^(4) = 17.32, *p* = 0.002), and hippocampal 5-HIAA (X^2^(4) = 14.08, *p* = 0.005), as well as hippocampal GSSG/GSH (X^2^(4) = 12.21, *p* = 0.02), GSH (X^2^(4) = 10.75, *p* = 0.03), and NA (X^2^(4) = 10.50, *p* = 0.03). Only cortical GSSG/GSH (Figure 5E; *p* = 0.05, d = 1.27 [0.2; 2.4]) was increased by ZEM25, whilst ESC administration increased cortical 5-HIAA (Table A4; *p* = 0.05, d = 1.74 [0.6; 3.0]) relative to UCMS placebo controls.

**Figure 5 cells-14-01029-f005:**
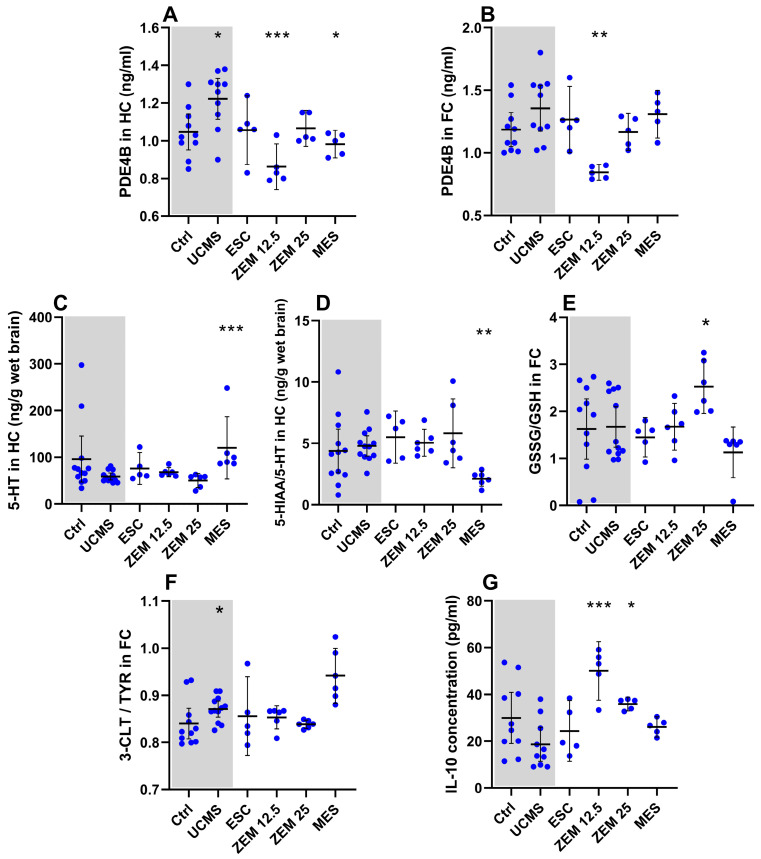
Treatment effects on neurochemical markers in male rats: effects of treatment on (**A**) PDE4B in the HC; (**B**) PDE4B in the FC; (**C**) 5-HT in the HC; (**D**) 5-HIAA/5HT ratio in the HC; (**E**) GSSG/GSH ratio in the FC; (**F**) 3-CLT/TYR ratio in the FC; (**G**) IL-10 in plasma. * *p* < 0.1, ** *p* < 0.01, *** *p* < 0.001. Data points represent the mean ± 95% CI. Statistical analyses are presented in-text and summarized in Table A4.

## 4. Discussion

The most important findings of this study are that the UCMS model induced anhedonia-like and hyperarousal (anxiety-like) behavior, reminiscent of anxious depression, although with worsened despair, in male UCMS rats. Importantly, the impact of UCMS on male and female rats differed, with a large effect observed in males but not female rats, suggesting sex-dependent stress sensitivity profiles (female treatment data are presented in Appendix A). Additionally, and regarding biomarkers related to MDD or anxiety, UCMS-exposed male rats showed a significant increase in hippocampal PDE4B and large effect size changes in cortical 5-HT (decreased) and DOPAC (increased), whilst these were unaltered in their female counterparts. ZEM12.5 and MES reversed UCMS-induced anhedonia-like behavior in male rats, with ZEM12.5 inducing sustained pro-hedonic effects. Compared to the UCMS group, ZEM12.5 decreased hippocampal and cortical PDE4B and increased plasma IL-10 concentrations. In turn, MES decreased hippocampal PDE4B concentrations, 5-HT turnover, and cortical GSSG, and increased hippocampal 5HT and DA relative to UCMS-exposed animals.

During model validation, UCMS failed to induce despair-associated immobility and/or reduced coping (swimming and climbing), in fact exacerbating it (UCMS groups; Figure 2B-D). While surprising, the UCMS model is known to be less reliable in inducing these behavioral stress responses [7] and where anhedonia has been noted as its primary MDD-like behavioral trait [47]. That said, other investigators [72,73] have questioned the translatability of the FST, suggesting that increased immobility may represent an adaptive response to conserve energy, while others believe that increased escape behavior is driven by anxiety [74]. This notion is supported by a decrease in immobility time in subsequent tests [75] and argues for behavior to be interpreted repeatedly and within the context of the overall experimental results [73]. Another option is to re-interpret the locomotor data as a read-out for MDD, anxiety, and anhedonia. This is the basis for the tripartite model [76,77].

Interpreting our findings within the context of the tripartite model of anxious depression considers the characterization of (1) decreased positive affect (i.e., increased anhedonia-like behavior), (2) increased negative affectivity (i.e., hyperlocomotor activity), and (3) hyperarousal [77]. The increased time spent in escape-directed behavior could then be the result of hyperlocomotor activity and even point to different coping strategies between male and female rats [78]. Considering the same increase in mobility in the FST observed in female rats (see above), this could indicate similar maladaptive stress responses indicative of anxiety based on the hyperarousal argument in male rats. The anhedonia-like effects of the UCMS may be masked in female rats due to the influence of ovarian hormones on taste in the SPT [79].

In the OFT, UCMS increased locomotor activity, independent of sex (UCMS M + F; Figure 2E). Separately, this increase was only present in male and not in female rats (UCMS M vs. UCMS F; Figure 2E; Table A1). Although a decrease in locomotor activity is most often representative of psychomotor retardation in depressed patients, the DSM-V also describes psychomotor agitation as a symptom of MDD [8]. Interestingly, Rantala and colleagues [80] suggest that psychomotor agitation can be driven by anxiety and could therefore be indicative of impaired adaptive responses. Alternatively, others [59,81,82] have ascribed such increased locomotor activity to hyperarousal. Hyperarousal is a common symptom of anxiety disorders [77], where even neutral or mild aversive stimuli can impair the arousal-inhibition system via the suppression of GABA neurotransmission, presenting as increased escape-directed behavior [81,83]. Moreover, exposure to white light (whether 600 or 25 lux) [60,84] and chronic single housing [85,86,87] can also induce hyperlocomotor activity and decreased immobility (due to increased swimming) in the FST. Since both factors were used in this study, these may have contributed to the observed increase in locomotor activity and subsequently the paradoxical effects observed in the FST (i.e., decreased immobility and increased swimming) (UCMS groups; Figure 2B–D), as noted above.

The lighting used, the presence of white noise, and having the platform elevated 90 cm above the ground prompt rodents to escape from the aversive open platform into a safe, dark space [88]. An anxious rat would spend more time in the center and inner zones of the BM while searching for the escape box, as opposed to exploring the riskier border and edge of the platform. In support of this, decreased outer/inner zone duration ratio in UCMS-exposed male (but not female) groups was evident (UCMS M vs. UCMS F; Figure 2F), confirming an anxious phenotype. The stress-exposed female rats did show a decrease in edge exploration characterized by head dips off the edge of the platform, which may indicate some anxious behavior. As for female rats, UCMS exposure only decreased edge exploration (head dips) in the BM (Table A1), indicating aversion to the edge, although no other parameters supported increased anxiety-like behavior. Interestingly, anxiety-like behavior was also not observed in the EPM in either male or female groups (Table A1). However, this could be due to certain methodological factors, such as conducting the EPM under red light and not more anxiogenic bright light [89,90] and/or the elevation difference between the BM (90 cm) and EPM (50 cm) platforms, where the former is arguably more anxiogenic [52,91].

Although anhedonia-like behavior is a recognized behavioral consequence of the UCMS protocol [47], this response was not evident when the sexes were combined. Despite the apparent differences in mean sucrose consumption over the experimental period (PND50 to 101), the change in sucrose consumption was comparable across all UCMS-exposed experimental groups. That said, earlier work [48] has described variable results regarding anhedonia following UCMS. However, when differentiating between sex, sucrose preference showed a statistically significant increase over time in UCMS-exposed female rats (*p* = 0.01; *R*^2^ = 0.10) (UCMS F; Figure 2A), whilst male rats showed a non-significant decrease (*p* = 0.46, *R*^2^ = 0.01). Nevertheless, as mentioned in Section 3.1.1, the UCMS-exposed male rats had a lower mean sucrose preference over the experimental period compared to control male rats and UCMS-exposed female rats. Although not statistically significant, the reduction in sucrose preference in males aligns with previous work [38,47,79,92], indicating sex-dependent anhedonia-like behavior in male rats. This data suggests that male rats are more sensitive to UCMS than their female counterparts. The impact of ovarian hormones on taste is thought to underlie the irregular sucrose preference displayed by UCMS-exposed female rats [79]. Indeed, some investigators have questioned whether the SPT is an appropriate behavioral test for anhedonia in female rats [38,79].

Surprisingly, when ignoring sex differences, the UCMS model did not induce any behavioral deficits in the BM to indicate or suggest altered cognitive function. In this regard, primary latency to and duration in the probe zone of the BM are accepted behavioral measurements of cognitive function [55]. That said, UCMS-exposed male Wistar rats made fewer primary errors in finding the escape box, which suggests improved cognitive function (Table A1). This is an unexpected finding. To our knowledge, this has not been described before in the BM, although similar results were described by Gouirand and Matuszewich [93] using the Morris water maze, a cognitive test that operates along the same principles as the BM. The authors theorized that a chronic stress-induced increase in hippocampal volume may engender an initial improvement in spatial memory. No differences in cognition were observed in the female rats. That aside, the BM may also have relevance in the assessment of anxiety, as highlighted earlier. In conclusion, UCMS engenders sex-dependent behavioral deficits in male rats, in particular anhedonia and anxiety.

Plant extracts are well-known to display an array of possible pharmacological actions [94]. In vitro studies were the first to show that ST binds SERT and PDE4 [29]. These findings spurred speculation as to how these actions may underlie the antidepressant actions of ST, the foremost among these being PDE4 and SERT inhibition [29]. Indeed, the presence of SERT polymorphisms exemplifies the causal role of 5-HT in both conditions [95]. While in vivo animal studies have shown distinct antidepressant-like behavioral effects [31,61], these putative mechanisms of action have not been described in vivo.

We observed an increase in hippocampal PDE4B concentrations in UCMS-exposed male rats (UCMS M; Figure 3A) with a similar, albeit non-significant, cortical pattern (UCMS M; Figure 3B). Contrastingly, no differences in PDE4B concentrations were observed in either brain region in UCMS-exposed females or in the combined group. Increased PDE4 activity reduces cAMP-CREB signaling, which can in turn increase inflammation [24], decrease neurotrophic factor synthesis [96], and impair monoamine-mediated activation of the adenylate cyclase-cAMP pathway [97,98]. Decreased PDE4 protein concentrations, as described here, can thus have a profound effect on a diverse range of signal transduction pathways [98]. Proof of concept is that PDE4 inhibitors, such as rolipram, exert or amplify antidepressant and anxiolytic actions [99]. In this study, the said increase in PDE4 concentrations could be causally associated with the anhedonia- and anxiety-like behavior observed in UCMS-exposed male rats (by reducing monoamine signaling; see below). In fact, this hypothesis is supported by the moderate negative correlation between hippocampal PDE4 concentration and sucrose preference and other similar observations [100], along with the successful reversal of decreased sucrose preference with PDE4 inhibitors (i.e., etazolate [101] and rolipram [102]). This also aligns with the concept of sickness behavior, where inflammation can cause symptoms like anhedonia and fatigue [40].

As alluded to earlier, increased PDE4 concentrations can increase inflammatory cytokine release from microglia and other immune cells [16,24], thereby prompting an inflammatory state that contributes to depressive and anxiety-related symptoms [103,104]. Tyrosine (TYR), the precursor of DA and NA [105], can also be converted to 3-chlorotyrosine (3-CLT), particularly in a pro-inflammatory state, and is therefore considered an indicator of chronic inflammation and oxidative stress [106,107]. In an inflammatory state, myeloperoxidase oxidizes Cl^−^ and H_2_O_2_ (hydrogen peroxide) to hypochlorous acid (HOCl^−^), which can react with tyrosine to form 3-CLT [108]. The conversion of tyrosine to 3-CLT is dependent on both inflammation (myeloperoxidase activity) and oxidative stress. It is therefore noteworthy that UCMS increased cortical tyrosine turnover—although this was only statistically significant in the combined group (UCMS M + F; Figure 3C; *p* = 0.003, *d* = 0.77 [0.2; 1.4]). Still, that neither plasma inflammatory markers (IL-10 and TNFα) nor hippocampal or cortical redox state markers (GSH/GSSG) were statistically altered by UCMS (Table A2) questions the precise redox pathways involved, as well as the extent to which these pathways may be affected.

Monoaminergic activity is widely accepted to be decreased in MDD and anxiety [3,40]. Monoamines in the two brain regions analyzed here did indeed confirm sex-specific monoamine differences in UCMS-exposed rats (Table A2). Elaborating, UCMS decreased cortical 5-HT and increased DOPAC concentrations in male rats, without altering 5-HIAA or DA concentrations, nor serotonergic (5-HIAA/5-HT) or dopaminergic (DOPAC/DA) turnover. In contrast, UCMS-exposed female rats presented with decreased cortical DOPAC concentrations and 5-HIAA/5-HT (Table A2). Of note, UCMS had no effect on NA concentrations in any of the experimental groups. The literature does, however, suggest that UCMS does not induce major effects on NA [109], highlighting the specificity of the model for serotonergic and dopaminergic processes.

Decreased cortical 5-HT levels noted in the UCMS-exposed male rats (UCMS M; Figure 3D) coincide with the monoamine deficit hypothesis of MDD [16,82,110] and could have contributed to the observed anhedonia-like behavior (UCMS M; Figure 2A) [21]. That said, a decrease in 5-HT was observed in the absence of altered turnover (5-HIAA/5-HT; male rats; Table A2). This could be attributed to increased inflammation-mediated activation of the tryptophan–kynurenine pathway, which switches tryptophan metabolism towards kynurenine and away from 5-HT synthesis [21]. Indeed, indoleamine 2,3-dioxygenase, involved in kynurenine synthesis, is activated by inflammatory and oxidative states [23]. Moreover, we observed increased tyrosine turnover and increased PDE4B concentrations in UCMS-exposed rats (Table A2), confirming the earlier-mentioned hypothesis. Alternatively, the decrease in 5-HT may indicate the depletion of 5-HT stores by VMAT [111]. With isolation rearing being present in the UCMS paradigm, a decrease in cortical 5-HT levels has been associated with increased locomotor activity in socially isolated rats [86,87], supporting our observations. Given sex-dependent differences in UCMS-exposed females and increased 5-HT turnover observed with MDD [111,112], the decrease in 5-HT metabolism observed in female rats may be related to corticosteroid and sex hormone differences [38].

Finally, UCMS increased cortical DOPAC in male rats yet decreased said levels in females (UCMS M vs. UCMS F; Figure 3E). Increased cortical DOPAC has also been observed in a previous UCMS rodent study [85]. Because DA is involved in reward processing [16], an increase in its metabolite, DOPAC, may allude to suboptimal dopaminergic signaling to explain the anhedonia-like behavior evident in male rats. The decreased cortical DOPAC concentrations in UCMS-exposed females suggest the opposite, especially when considering the higher (albeit non-significant) mean sucrose consumption and significant increase over time.

In conclusion, UCMS is a translational model of MDD and anxiety that presents with deficits in serotonergic and dopaminergic processes that are most prominent in males. Moreover, oxidative stress (increased tyrosine turnover) and possible neuromodulation and inflammation (increased PDE4B) are comorbid pathologies.

None of the UCMS-altered bio-behavioral parameters were reversed by the reference antidepressant, ESC (Table A3). Nevertheless, ESC increased sucrose preference over time, albeit non-significantly, providing some predictive validity for the UCMS model. That said, anhedonia is notoriously difficult to treat with first-line antidepressants, such as ESC and venlafaxine [113,114]. Therefore, our findings are somewhat reflective of clinical findings. Surprisingly, ESC was also unable to improve anxiety-like behavior and related changes in neurochemical markers, especially considering its SERT inhibitory actions. Instead, ESC induced statistically significant anxiogenic-like behavior in UCMS-exposed male rats, including thigmotaxis in the OFT (ESC; Figure 4E) and decreased edge exploration in the BM (ESC; Figure 4F). The dose-dependent effects of ESC may be relevant here and require further investigation [115]. However, increased cortical 5-HIAA following ESC treatment may indicate decreased 5-HT activity, which could contribute to depressive-like symptoms [116] but is also indicative of antidepressant effects, especially since decreased cerebrospinal fluid 5-HIAA concentrations have been associated with MDD [3]. Such conflicting results can thus complicate the interpretation of our findings, especially since there were no other informative serotonergic differences.

Unlike ESC, both ZEM12.5 and MES administrations were effective in reversing some of the UCMS-induced bio-behavioral alterations. Oddly, while MES reversed UCMS-induced anhedonia-like behavior in male rats, this effect was transient, in contrast to the sustained pro-hedonic-like effects of ZEM12.5. ZEM12.5 decreased hippocampal and cortical PDE4B and increased plasma IL-10 concentrations versus the UCMS control group. In turn, MES decreased hippocampal PDE4B concentrations, 5-HT turnover, and cortical GSSG, and increased hippocampal 5HT and DA relative to UCMS control animals. In fact, when considering the overall effect of these treatments on all bio-behavioral parameters, MES (*d* = 0.81 [0.6; 1.0], CoV = 72%) induced a larger and more consistent effect than that observed in the ZEM12.5 group (*d* = 0.64 [0.4; 0.9], CoV = 97%). This supports the idea that MES is a significant alkaloid constituent of ST, being attributed with serotonergic, anti-inflammatory, PDE4 inhibitory, and cytoprotective properties [28]. Of note, our previous work in zebrafish larvae [31] indicates that the most effective anxiolytic- and antidepressant-like doses, respectively, correlate with ZEM12.5 and ZEM 25 in rats. To this end, MES appeared to only induce a transient increase in sucrose preference in UCMS-exposed male Wistar rats, whilst ZEM12.5-administered rats displayed an overall increase in sucrose preference across the experimental period (MES and ZEM12.5; Figure 4A), thereby highlighting the pharmacodynamic synergy of the standardized extract.

Considering the negative correlation between hippocampal PDE4B levels and anhedonia-like behavior in the UCMS-exposed male rats, the ZEM12.5-induced decrease in PDE4B concentrations (ZEM12.5; Figure 5A,B) may therefore directly contribute to the observed improvement in sucrose preference in these animals. Conversely, the transient improvement in anhedonia displayed by the MES-administered male rats is supported by the smaller attenuating effect on hippocampal PDE4B concentrations and is suggestive of other alkaloids being responsible for the larger effect observed with ZEM12.5 [31].

The reduction in general locomotor activity in the BM following administration of ZEM12.5 to male UCMS rats (yet not significant in the OFT) could indicate anxiolytic-like effects not observed at the higher dose (i.e., ZEM25), further illustrating the dose-dependent effects of ZEM [28,61] (Table A3). However, the EPM (Table A3) was unable to confirm any additional anxiety-related changes. That said, the EPM and the BM evaluate different aspects of anxiety-like behavior, namely hyperarousal (increased locomotor activity) versus an increase in the natural agoraphobic response (threat avoidance), respectively [52]. Also, keeping in mind that the hyperarousal response could increase mobility in the FST as discussed earlier, ZEM12.5 decreased swimming, thereby reversing the effects of UCMS (ZEM12.5; Figure 4C). This can also be related to reduced PDE4B levels, which would normalize 5-HT-mediated subcellular cAMP signaling via restored 5-HT_1A_ activation. In fact, this is a well-known antidepressant and anxiolytic action [3]. Either way, ST is known to possess strong serotonergic properties [28], which comes across in this work. It is therefore noteworthy that MES increased hippocampal 5-HT (MES; Figure 5C) and decreased 5-HT turnover (MES; Figure 5D), most likely via modifying synthesis and/or release (VMAT2 upregulation or SERT inhibition [28]), and reduced metabolism (via MAO inhibition [117]). It is important to consider that UCMS only altered cortical and not hippocampal 5-HT concentrations, while MES selectively increased 5-HT in the hippocampus. This contradiction warrants further investigation. MES also increased cortical DA, which can contribute to its transient antianhedonic effects in the SPT.

None of the interventions were effective in reversing the UCMS-induced increase in cortical tyrosine to 3-CLT turnover (all treatment groups versus UCMS; Figure 5F), suggesting minimal antioxidant and/or anti-inflammatory effects in vivo. Interestingly, although not statistically significant, MES did trend toward increased cortical 3-CLT/TYR, hence a putative prooxidant effect that would counteract the anticipated therapeutic effect. In fact, this contrasts with the in vitro findings of Bennett and Smith [118], who reported antioxidant and anti-inflammatory effects with a high-MES extract. Regarding dose, the dose of MES correlates with ZEM12.5, which also showed no effect on cortical or hippocampal 3-CLT/TYR values. Yet, MES decreased cortical GSSG (Table A4), indicating antioxidant potential. Considering the opposing effects of increased 3-CLT/TYR but decreased GSSG, one might argue that the mechanism of MES pivots toward antioxidant properties rather than anti-inflammatory properties, at least in this context. In contrast, albeit not significant, ZEM25 did show a trend towards decreased 3-CLT/TYR (*p* = 0.17, *d =* 1.35 [2.5; 0.3]) (Table A4), indicating at least some anti-inflammatory and/or antioxidant activity. That said, dose-dependent effects of the extract or the ratio of constituents present may be responsible. However, several antioxidant and/or anti-inflammatory plant extracts are active in vitro yet lose this activity when assessed in vivo, possibly due to pharmacokinetic influences such as tissue bioavailability and stability in the gastrointestinal tract [119].

However, ZEM25 decreased GSH and increased GSSG/GSH in the frontal cortex (ZEM25; Figure 5E), indicating prooxidant effects but, at the same time, profound anti-inflammatory actions via increases in IL-10 (ZEM25; Figure 5G). Although these simultaneous effects seem counterintuitive, these specific markers work on different mechanisms and thus are very much dependent on cellular redox and inflammatory state. For example, antioxidants can present with both anti- and prooxidant actions, which vary depending on the in vivo redox status of the cell as well as the neuro-progressive state of the condition itself [120,121]. Therefore, the redox state of the UCMS model may differ in complexity compared to the pathophysiological aspects of a mood and anxiety disorder and in any given patient. ZEM25, therefore, appears to have opposing actions regarding redox-inflammation markers, but it also displays the opposite effect to MES, presenting with a predilection for anti-inflammatory rather than antioxidant activity. In this regard, the pharmacodynamic interaction between the ST constituents may not necessarily be synergistic. In fact, the observed MES-induced antioxidant and ZEM25-induced anti-inflammatory actions may be counterbalanced by other ST alkaloids—an interesting hypothesis to consider.

Lastly, despite no effects of UCMS on plasma IL-10 levels, ZEM12.5 increased this anti-inflammatory cytokine in UCMS-exposed male rats (ZEM12.5; Figure 5G), which confirms the in vitro findings of Bennett and Smith [118]. This action will further contribute to the therapeutic effects of ZEM12.5 discussed above. Importantly, this was not seen with MES, again indicating that another alkaloid or other synergistic effects may underlie the anti-inflammatory action of the extract.

Some limitations need to be noted. The eventual focus on males resulted in a smaller sample size than planned, although it did not compromise the statistical and ethical grounding of the study (Appendix A). The measurement of female hormones and corticosterone, assessing PDE4B activity (not expression) as well as assessing receptor activity related to the proposed mechanisms of ST, could have provided a clearer picture. The complexity of plant extracts blurs the interpretation of bio-behavioral data due to possible interactions between ingredients. Finally, the lack of inter-laboratory reliability of the UCMS model complicates repeatability and interpretations of findings across studies [48].

## 5. Conclusions and Unifying Hypothesis

Male UCMS-exposed rats present with anxiety and anhedonia, as well as deficits in serotonergic and dopaminergic processes, and neuro-inflammation (increased PDE4B). Effects on cognition were negligible in this study, although various studies indicate pro-cognitive effects, which suggests that the lack of effects may be attributed to overtraining in the BM. Female rats show resilience to UCMS. The biological response to UCMS is summarized in Figure 6, while also describing the neurochemical processes engaged by ZEM12.5 and MES to elicit an antidepressant- and anxiolytic-like effect. ZEM12.5 displays anxiolytic and pro-hedonic effects, along with potent PDE4B inhibition and anti-inflammatory effects (increased IL-10). MES displays transient pro-hedonic effects, alongside PDE4B inhibition, monoaminergic properties, and antioxidant effects. We propose that synergistic mechanisms exist between MES and other ST alkaloids, exemplified by potent PDE4B inhibition with ZEM12.5. Moreover, other alkaloids present in the extract may antagonize the monoaminergic and antioxidant properties of MES and thus explain its stronger neurochemical effects when administered alone versus the ZEM12.5 extract. In conclusion, multitargeted actions on monoamines, redox-inflammatory, and PDE4 provide ST with antidepressant effects across multiple symptom domains, thus offering unique therapeutic opportunities in the treatment of MDD.

## Figures and Tables

**Figure 1 cells-14-01029-f001:**
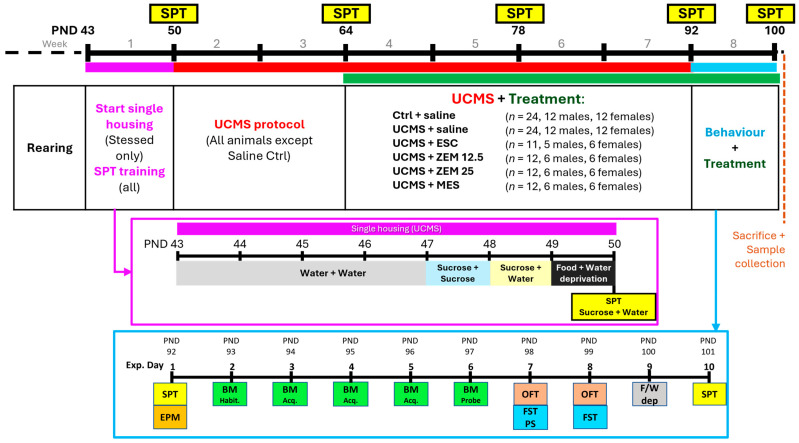
Study layout, showing the general timeline and experimental groups (black box), the timeline of the initial week of SPT training and single housing (purple box), and the behavioral test week (blue box). The broad colored lines at the top represent the various exposure periods: Purple—single housing and sucrose preference training; red—UCMS exposure (all groups except Ctrl animals); green—treatment period; blue—behavioral testing. BM Habit.—Barnes maze habituation; BM Acq.—Barnes maze acquisition; BM Probe—Barnes maze probe trial; EPM—elevated plus maze; FST PS—forced swim test pre-swim; FST—forced swim test; F/W dep—food and water deprivation; OFT—open field; PND—postnatal day; SPT—sucrose preference test; UCMS—unpredictable chronic mild stress.

**Figure 2 cells-14-01029-f002:**
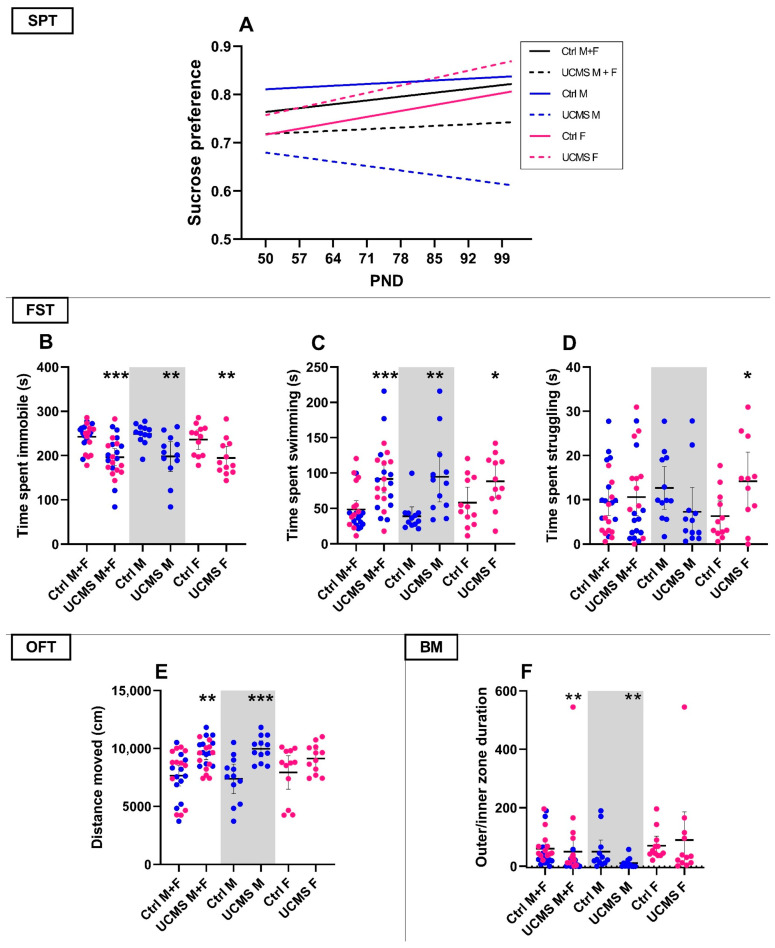
Behavioral tests for model validation in rats: (**A**) sucrose preference over time comparing the control (solid lines) and UCMS (stippled lines) groups of the combined sex group (black) compared to male (blue) and female (pink) rats. PND—postnatal day. B–F compares the Ctrl and UCMS groups of the combined sex group presented next to those of male and female rats alone. (**B**) Immobility, (**C**) swimming, (**D**) struggling behavior in the forced swim test; (**E**) distance moved (cm) in the OFT. (**F**) Time spent in the outer/inner zones of the BM. Data points represent the mean ± 95% CI, with male and female rats indicated in blue and pink, respectively. Statistical analyses are presented in-text and summarized in Table A1 in Appendix B. Statistical significance (**p* < 0.05, ** *p* < 0.01, *** *p* < 0.001) is indicated above the UCMS group of the specific validation group.

**Figure 6 cells-14-01029-f006:**
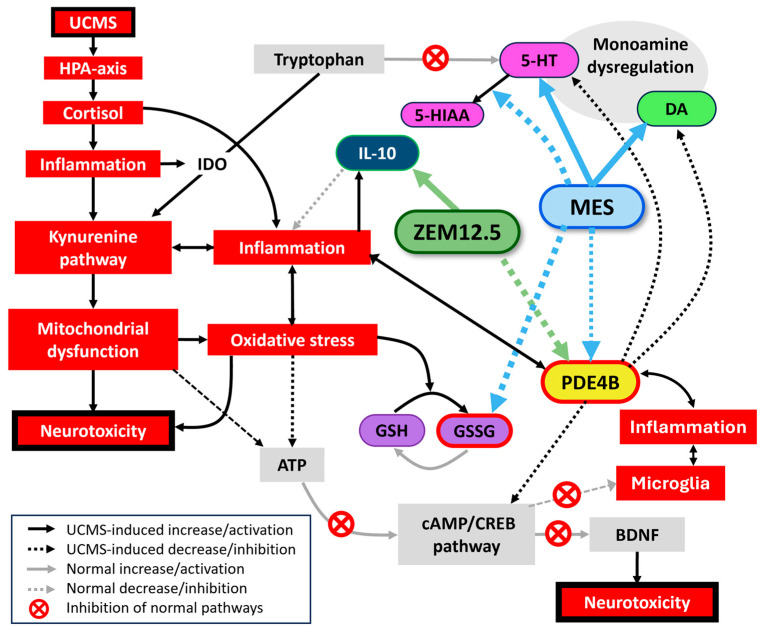
Summary of how ZEM12.5 and MES may induce antidepressant and anxiolytic actions and their points of synergy. UCMS chronically activates the HPA axis, leading to hypercortisolemia release and a pro-inflammatory state. Inflammatory cytokines increase IDO, shifting tryptophan metabolism in favor of the kynurenine pathway, consequently decreasing tryptophan availability for 5-HT production. The latter leads to monoamine dysregulation with MDD and/or anxiety as a result. Disordered kynurenine metabolism leads to quinolinic acid release, mitochondrial dysfunction, oxidative stress, and neurotoxicity. In turn, oxidative stress is pro-inflammatory. These two processes form a cycle that amplifies cellular damage. Oxidative stress oxidizes glutathione, increasing GSSG and decreasing GSH, while also decreasing the energy production (ATP) necessary for the cAMP/CREB cycle. Since increased PDE4B and reduced cAMP increase pro-inflammatory cytokine release, normal cAMP-mediated suppression of microglial activation is inhibited, leading to decreased BNDF, decreased neuroplasticity, pro-inflammatory cytokine release, and a pro-inflammatory/neurotoxic state. Inflammation also increases PDE4B expression to hasten cAMP metabolism, resulting in dysregulated monoamine (5-HT, DA) signaling and depressive symptoms (anhedonia, anxiety, etc.). ZEM12.5 decreases cortico-hippocampal PDE4B and increases plasma IL-10, inducing anti-inflammatory and neuroprotective effects. MES is anti-inflammatory by decreasing hippocampal PDE4B, albeit to a lesser degree than ZEM12.5. MES increases hippocampal 5-HT and cortical DA and decreases hippocampal 5-HT turnover, thereby improving monoamine functioning, as well as decreases cortical GSSG with resulting antioxidant effects. The absence of the monoaminergic and antioxidant effects of MES in ZEM12.5 is unclear but could reside in antagonistic effects by other alkaloids. 5-HT—serotonin; 5-HIAA—5-hydroxyindoleacetic acid; ATP—adenosine triphosphate; BDNF—brain-derived neurotrophic factor; cAMP/CREB—cyclic adenosine monophosphate (cAMP)/cAMP response element-binding protein (CREB) pathway; DA—dopamine; HPA-axis—hypothalamic–pituitary–adrenal axis; IDO—indoleamine 2,3-dioxygenase; GSH—reduced glutathione; GSSG—oxidized glutathione; IL-10—interleukin 10; MES—mesembrine; PDE4B—phosphodiesterase 4B; Zembrin^®^ 12.5 mg/kg.

## Data Availability

All relevant data sets are included in this article, Appendix B, and the accompanying Appendix A. Any additional information is available upon request from the authors.

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
