# Peer review of "Neuro-Inflammatory and Behavioral Changes Are Selectively Reversed by *Sceletium tortuosum* (Zembrin^®^) and Mesembrine in Male Rats Subjected to Unpredictable Chronic Mild Stress"

_cells, 2025, doi:10.3390/cells14131029_

Round 1
Reviewer 1 Report
Comments and Suggestions for Authors
This manuscript appears to be part of a thesis. The descriptions of studies should be carefully revised prior to submission. The work of this manuscript focuses primarily on behavioral and phenotypic outcomes, with limited investigation into underlying mechanisms. Figure 6 should be redrawn, as it suggests that PDE4B expression increases directly in response to UCMS; this interpretation requires clarification. Additionally, the authors should discuss how increased PDE4B expression might contribute to mitochondrial dysfunction and neuroinflammation, to provide a more mechanistic understanding of the findings.
Comments on the Quality of English LanguagePlease clarified sentences in the introduction, materials and methods, results and discussion.
Author Response
REFEREE 1:
Comment 1: Considering the first referee report, the referee indicates that all the following "Must be improved":
- Does the introduction provide sufficient background and include all relevant references?
- Is the research design appropriate?
- Are the methods adequately described?
- Are the results clearly presented?
- Are the conclusions supported by the results?
These comments suggest strong reservations across the board, although the referee unfortunately does not support these criticisms with detail taken from the paper. Instead, the referee emphasizes that the description of the study should be carefully revised prior to submission. In line with these comments, we have re-assessed the manuscript from beginning to end, now completely revising the Methods, Results, Discussion and Conclusion sections, as well as the figures and the legends. This did indeed highlight shortfalls that we believe will address the referee’s concerns. However, if this does not suffice, please could the referee clarify what aspects should be edited with specific examples?
Comment 2: The work of this manuscript focuses primarily on behavioral and phenotypic outcomes, with limited investigation into underlying mechanisms.
This is incorrect. The paper examines several biomarkers relevant to depression, especially those focusing on neurotransmission and neuroinflammation. Thus, the following biomarkers were analyzed: cortico-hippocampal monoamines such as dopamine, serotonin and their metabolites, inflammatory markers such as phosphodiesterase (PDE) 4B, interleukin (IL)-10, and tumor necrosis factor (TNF)-α, and redox markers such as tyrosine, 3-chlorotyrosine, and reduced and oxidized glutathione (GSH and GSSG). We then elaborate extensively in the revised Discussion how these biological changes correlate with the observed behavioral changes described in the Results to provide a holistic mechanistic understanding of how behavior and biology interact in the treatment-naïve UCMS model. Thereafter, we explain how the standardized Sceletium tortuosum extract and/or mesembine addresses the described bio-behavioral deficits evident in the UCMS model. Finally, we interface this detail with how these actions may be of translational value in treating depression.
Comment 3: Figure 6 should be redrawn, as it suggests that PDE4B expression increases directly in response to UCMS; this interpretation requires clarification.
Thank you for this important comment. After studying Figure 6, it is indeed clear why the referee found the figure misleading and unclear regarding the link between UCMS and PDE4B expression. We have now completely revised Figure 6, as requested. We also provide a detailed legend explaining how UCMS eventually links with PDE4B expression. Briefly, UCMS induced chronic stress and hypercortisolemia induces inflammation which increases PDE4B expression via a series of interlinking steps, as shown in the figure. We trust that these improvements will address the referee’s concerns. Please refer to Figure 6 and its legend on page 23 of the revised manuscript.
Comment 4: Additionally, the authors should discuss how increased PDE4B expression might contribute to mitochondrial dysfunction and neuroinflammation, to provide a more mechanistic understanding of the findings.
Our apologies, it is understandable that Figure 6 would not have been adequate in describing how increased PDE4B expression might contribute to mitochondrial dysfunction and neuroinflammation. Just to be clear, PDE4B is not immediately involved in mitochondrial function, while our study did not directly look at that. Based on our data, we suggest that increased PDE4B expression is primarily driven by stress-induced inflammation. Inflammation and oxidative stress (and mitochondrial dysfunction) mutually reinforce one another, creating a cycle that amplifies tissue damage over time. Cyclic AMP primarily acts to dampen microglial activation to reduce the release of pro-inflammatory cytokines, promoting a shift towards an anti-inflammatory state. Thus, increased PDE4B, decreased cAMP, and oxidative stress are indirectly related via their proinflammatory actions, as is evident in the revised Figure 6. Viewing the broader picture, increased PDE4B expression is a product of a chronic stressor, such as UCMS, that drives neuroinflammation by increasing the release of proinflammatory cytokines. Thus, PDE4B inhibition will counter neurotoxicity and neuroinflammation (bottom right of revised Figure 6) as well as rejuvenate monoamine signaling (top right of revised Figure 6). By virtue of their PDE4 inhibitory actions, this is the predicted mechanism underlying ZEM and MES administration.
To address this mechanism more clearly, we have significantly revised Figure 6. In addition, we provide a detailed legend explaining how UCMS induced neuroinflammation modulates PDE4B, but is also modulated by PDE4B. We trust that these improvements will address the referee’s concerns. Please refer to revised Figure 6 and its legend on page 23 of the revised manuscript.
Comment 5: Please clarify sentences in the introduction, materials and methods, results and discussion.
This comment essentially refers to the whole manuscript. As noted in our earlier responses and significant amendments to the paper, we trust that we have now addressed these shortfalls.
Comment regarding English language:
The referee indicates that English could be improved to more clearly express the research. The referee unfortunately does not provide details on where any language editing is needed. Our paper went through robust language editing prior to submission, so we are uncertain what more can be done. Nevertheless, we have re-assessed the entire manuscript from beginning to end, subjecting it to a thorough language edit, and significant improvements have been made. We trust this will meet with the referee's requirements. However, if this does not suffice, please could the referee clarify what aspects should be edited with specific examples.
Reviewer 2 Report
Comments and Suggestions for Authors
This interesting describes in vivo evaluation of the antidepressant and anxiolytic properties of Sceletium tortuosum (ST) and its principal alkaloid, mesembrine, in a unpredictable chronic mild stress in rats. The authors effectively compare ST extract at two doses (ZEM25 and ZEM12.5), mesembrine, and escitalopram, using a well-rounded behavioral and biochemical analysis. Additionally, the study shows sex-specific responses, revealing that male rats have notable behavioral and neurochemical alterations. The findings suggest that ZEM12.5 and mesembrine affect mood-related behaviors and neuroinflammation, perhaps via PDE4B inhibition and monoaminergic regulation. This work provides significant insights into the therapeutic potential and intricacies of ST, notably highlighting the dose-dependent and potentially synergistic effects of its constituents.
I have a couple of comments if the authors could improve the readability of the tables by marking significance. The figure descriptions should include to which group the test groups are compared, e.g. ZEM 12.5 in Figure 4B. Please add figures for the other behavioral tests: EPM, OFT and BM, both in the validation and in the section where substances are administrated.
In general, the paper is well written and I recommend it for publication.
Author Response
Comment 1: The authors could improve the readability of the tables by marking significance.
Thank you for this comment. We did in fact indicate statistical significance (p- and d-values) in bold in the tables, although this was not noted in the table legends. We have included this for clarification (see changes highlighted in yellow in the new table legends).
Comment 2: The figure descriptions should include to which group the test groups are compared, e.g. ZEM 12.5 in Figure 4B.
Thank you for this comment. We have followed this advice when referring to figures in the Discussion. However, the different treatments and comparisons between groups are specifically stated in-text, and since addition of the groups along with the figure reference is redundant, we have not done this in the Results section. We trust this is acceptable. Please see these changes throughout the Discussion, highlighted in yellow.
Comment 3: Please add figures for the other behavioral tests: EPM, OFT and BM, both in the validation and in the section where substances are administrated.
The figures for the OFT and BM are included in Figures 2 and 4, although the EPM data was intentionally omitted due to a lack of significant findings, and to shorten the already lengthy article. This is especially true for the validation data. For reasons of transparency, all data are nonetheless included in the tables, with only the most notable data presented in the figures. We have noted this fact in the revised manuscript (see lines 402-405). Therefore, no EPM data was presented in figures since there were no observable differences in either the validation or treatment experiments, but were included in the tables for transparency. However, we have added the time spent in the corners of the OFT, and edge exploration and distance moved in the BM in Figure 4(E-G), in the treatment response sections. This was done since we do mention these findings in the text, as well as include reference to these figures in the text. Where these excluded data are referenced to the Appendix, please see lines 408-413, lines 425-426, lines 444-445,462-464, lines 478-479, lines 488-489, line 498, lines 505 and 513, line 533-534, lines 556-558, line 564, and line 571.
Round 2
Reviewer 1 Report
Comments and Suggestions for Authors
The authors have revised the manuscript. I have no further comments
Author Response
The referee has indicated agreement with our revised manuscript, and has no further comments. The referee also indicates that the English has been improved and does not require further improvement.